# Tick-Derived Peptide Blocks Potassium Channel TREK-1

**DOI:** 10.3390/ijms25158377

**Published:** 2024-07-31

**Authors:** Canwei Du, Linyan Chen, Guohao Liu, Fuchu Yuan, Zheyang Zhang, Mingqiang Rong, Guoxiang Mo, Changjun Liu

**Affiliations:** 1School of Life and Health Sciences, Hunan University of Science and Technology, Xiangtan 411201, China; 2College of Life Sciences, Nanjing Agricultural University, Nanjing 210095, China; 3National & Local Joint Engineering Laboratory of Animal Peptide Drug Development, College of Life Sciences, Hunan Normal University, Changsha 410006, China; 4Frontiers Medical Center, Tianfu Jincheng Laboratory, No. 387-201 Heming Street, Chengdu 610212, China

**Keywords:** ticks, TREK-1 channel, IstTx, inhibition, acidification

## Abstract

Ticks transmit a variety of pathogens, including rickettsia and viruses, when they feed on blood, afflicting humans and other animals. Bioactive components acting on inflammation, coagulation, and the immune system were reported to facilitate ticks’ ability to suck blood and transmit tick-borne diseases. In this study, a novel peptide, IstTx, from an *Ixodes scapularis* cDNA library was analyzed. The peptide IstTx, obtained by recombinant expression and purification, selectively inhibited a potassium channel, TREK-1, in a dose-dependent manner, with an IC_50_ of 23.46 ± 0.22 μM. The peptide IstTx exhibited different characteristics from fluoxetine, and the possible interaction of the peptide IstTx binding to the channel was explored by molecular docking. Notably, extracellular acidification raised its inhibitory efficacy on the TREK-1 channel. Our results found that the tick-derived peptide IstTx blocked the TREK-1 channel and provided a novel tool acting on the potassium channel.

## 1. Introduction

Ticks (suborder Ixodida) are external parasitic arachnids that live by feeding on the blood of mammals, birds, and sometimes amphibians [1]. Most ticks are classified into two major families, the hard ticks and the soft ticks. *Nuttalliella namaqua*, found in southern Africa, is the only tick of family Nuttalliellidae, and it displays different characteristics, such as the position of the stigmata, from two other families [2]. Ticks transmit a number of infections caused by various pathogens, including rickettsia and other types of bacteria, viruses, and protozoa, when they bite their hosts. Tick-borne diseases cause substantial economic losses, estimated to affect ~80% of cattle worldwide [3]. There are increasing reports that tick-borne diseases occur in humans, such as Lyme disease and query fever [4,5,6]. Some species, notably the Australian paralysis tick, are also intrinsically venomous and can cause tick paralysis [7]. The salivary glands of ticks contain a variety of bioactive components that inhibit inflammation, clotting, and immune response, which facilitate ticks’ feeding on blood and the transmission of tick-borne pathogens [8,9,10]. For example, Salp15, isolated from the salivary glands of *Ixodes scapularis*, plays a critical role in the pathogen transmission of Lyme disease, and Salp15 can inhibit the activation of CD4^+^ T lymphocytes, reduce the production of IL-2, and inhibit the activation and proliferation of T cells, thereby inhibiting the immune stress response of the host [11,12]. Studying the bioactive components in the salivary glands of ticks and in the corresponding targets is of great significance for preventing ticks from biting and blocking the transmission of tick-borne pathogens [13,14,15].

Two-pore domain potassium (K2P) channels are important to control the resting membrane potential in almost all cells, known as the leaky potassium channel [16]. Fifteen distinctive K2P channels, including the first tandem of P domains in a weak inward rectifying potassium (TWIK) channel, have been identified in humans, and the family structurally contains four transmembrane segments and has active dimers [17]. The TWIK-related potassium (TREK) channel-1 is a mechano- and thermo-gated channel, and the channel can be regulated in various ways, such as through anionic lipids and cell swelling [18]. The TREK-1 channel mediates potassium in a strong voltage-dependent manner due to an ion-flux gating mechanism [19], and the TREK-1 channel also exhibits outward rectification because of an external magnesium block at negative potential [20]. The TREK-1 channel is expressed in various tissues and organs, including the sensory neurons of the dorsal root and trigeminal ganglia in the nervous system, and the heart and pulmonary arteries in the cardiovascular system [21,22]. Volatile anesthetics, such as chloroform and sevoflurane, can activate the TREK-1 channel on astrocytes to reduce membrane potential and inhibit neuronal excitability [23]. TREK-1 channel knockout mice have displayed less sensitivity to anesthetics [24,25]. The link between the TREK-1 channel and depression is demonstrated by TREK-1 channel knockout mice [26]. The persistent opening of the TREK-1 channel will induce the prolongation of the cell resting potential, reduce the frequency of action potentials, and lead to a decrease of serotonin synthesis, which cause depression [27]. On the other hand, the hypothalamus–pituitary–adrenal hormone axis (HPA) is overexcited in depression, which results in excessive secretion of glucocorticoids, induces upregulation of the TREK-1 channel, and thus inhibits the proliferation and differentiation of neurons [28,29].

In this study, to analyze the bioactive peptides in the salivary gland of *Ixodes scapularis* and the latent targets, peptide IstTx was explored, with its sequence obtained from the tick cDNA library. The peptide IstTx, by recombinant expression, exhibited an inhibitory efficacy on the TREK-1 channel, but not on the others. The peptide displayed different mechanisms from fluoxetine binding to the TREK-1 channel, which was hinted at by channel mutagenesis and molecular docking. Moreover, extracellular acidification could increase the inhibitory activity of IstTx on the TREK-1 channel. Our finding indicated that the tick-derived peptide IstTx blocked the TREK-1 channel in a concentration-dependent manner, and unique features of the peptide acting on the TREK-1 channel suggested IstTx may be a novel tool for exploring the physical and physiological effect of the TREK-1 channel.

## 2. Results

### 2.1. Recombinant Expression and Purification of the Peptide IstTx

Due to limitations in the collection and purification of animal venom, a cDNA library was built to analyze the peptide resource of *Ixodes scapularis* [30]. From the cDNA library, a sequence encoding a 108-residue peptide attracted our interest. The mature peptide, named IstTx, contains 89 residues with a molecular weight (MW) of 9854.9 Da when its signal peptide is released, which is predicated by SignalP 5.0 (https://services.healthtech.dtu.dk/service.php?SignalP accessed on 27 June 2024) (Figure 1A). IstTx displayed a high homology with peptides from other ticks, in particular, 92.13% identified with complement inhibitor CirpT4 from *Amblyomma americanum*, 72.94% identified with CirpT3 from *Rhipicephalus sanguineus*, and IstTx also exhibited a similar structure to the venom peptide MmKTx1 and U-scoloptoxin (16)-Sm2a (Sm2a) from *Ixodes scapularis*, including eight cysteines at the same position (Figure 1B). Using Swiss modeling, the three-dimensional (3D) structure of IstTx was predicted based on the data of CirpT1, the crystal structure of which has been resolved [31]. A 3D structure with a GMQE (global model quality estimate) of 0.76 showed that IstTx overlapped highly with CirpT1, and IstTx was fastened by four intramolecular disulfide bonds (Figure 1C).

To explore the function of IstTx, we constructed a plasmid of pET-32a (+) expressing recombinant IstTx in *E. coli* BL21 (DE3). As shown in Figure 2A, the DNA sequence encoding the gene of the IstTx was inserted into pET-32a (+) at the 3’ end of the fusion tag, which contains the trx sequence, 6×His tag, and enterokinase recognition site, sequentially. After expression, the fusion protein was purified by a Ni-NTA column (Figure 2B) and digested by enterokinase to release IstTx (Figure 2C), which was isolated by gel filtration on Sephadex G-75 (Figure 2D). To further purify the peptide, reversed-phase high-performance liquid chromatography (RP-HPLC) was performed (Figure 2E). The purified IstTx displayed a MW of 9848.5 Da (Figure 2F), suggesting that it contained four intramolecular disulfide bonds. Thus, we identified a novel peptide from *Ixodes scapularis* by cDNA library screening and obtained IstTx via recombinant expression and multiple purifications.

### 2.2. IstTx Inhibited the TREK-1 Channel

Patch clamp recording was performed to check the effect of IstTx on ion channels. When the current of the TREK-1 channel was induced by a membrane potential of +80 mV, IstTx displayed inhibitory efficacy on the channel (Figure 3A). As shown in Figure 3B, a voltage ramp recording also showed that IstTx inhibited the TREK-1 channel, and notably, the current evoked from the TREK-1 channel was almost abolished when perfused with 50 μM IstTx. We analyzed the inhibition and found that IstTx inhibited the TREK-1 channel in a concentration-dependent manner, with an IC_50_ of 23.46 ± 0.22 μM (Figure 3C). Following that, we checked the effect of IstTx on other K2P channels. For the current of the TREK-2 channel induced by a voltage ramp recording, IstTx at 50 μM exhibited little activity (Figure 3D,H). As shown in Figure 3E–H, IstTx at 50 μM slightly inhibited currents of the TASK-1, TASK-3, and TRESK channels elicited by a membrane potential of +80 mV, suggesting that IstTx selectively inhibited the TREK-1 channel. Therefore, we found that the peptide IstTx could inhibit the TREK-1 channel.

### 2.3. Selectivity of IstTx

To explore the selectivity of IstTx, we checked the effect of IstTx on some other channels, such as voltage-gated sodium channels (Nav) and voltage-gated potassium channels (Kv). The peptide IstTx did not show any inhibition on Nav1.3, Nav1.4, Nav1.5, and Nav1.7 (Figure 4A–D) or on the potassium channels, including Kv1.5, Kv2.1, Kv4.1, and the BK channel (Figure 4E–H). In addition, IstTx did not exhibit any effect on the voltage-gated calcium channel (Cav3.3), the calcium-activated chloride channel (Ano1), or the proton-activated proton channel (Otop1) (Figure 4I–K). The above results indicate that IstTx selectively inhibited the TREK-1 channel.

### 2.4. Some Features of IstTx Inhibiting the TREK-1 Channel

A stepwise voltage protocol was applied to check the voltage dependence of IstTx in inhibiting the TREK-1 channel. Similar to the result of the voltage ramp recording (Figure 3A), IstTx inhibited the TREK-1 channel in a voltage-independent manner when the channel was activated stably by different membrane potentials (Figure 5A,B). Following that, we analyzed the kinetics of IstTx binding to the TREK-1 channel when evoked by +80 mV. The curve showed that IstTx could inhibit the TREK-1 channel at a rapid rate and could also be washed off from the channel quickly (Figure 5C), which is different from the kinetics of fluoxetine inhibiting the TREK-1 channel [29]. The TREK-1 channel is regulated by multiple mechanisms, including activation by arachidonic acid (AA) [32]. Prior to this, spadin was reported to inhibit the TREK-1 channel in a direct manner, but could not block the channel when pre-activated with 10 μM AA [33]. We found that 10 μM AA evoked a large current on cells transiently transfected of the TREK-1 channel when the channel was elicited by a ramp recording of −100 to +40 mV. IstTx at 25 μM inhibited almost half the current of the TREK-1 channel when the channel was pre-stimulated by AA (Figure 5D), and the inhibitory efficacy was similar to that of the channel when activated by a voltage without AA, suggesting that IstTx may inhibit the TREK-1 channel via a different mechanism.

### 2.5. Acidification Enhanced the Inhibition of IstTx on the TREK-1 Channel

Protons were reported to inhibit the TREK-1 channel by reducing the open possibility of the channel [34]. We also proved that low pH could reduce the current of the TREK-1 channel, with a pKa of 7.29 (Figure 6A,B). The efficacy of some animal toxins on their corresponding receptors was strongly potentiated in an acidic environment [35]. The question remains whether extracellular acidification enhances the effect of IstTx acting on the TREK-1 channel or not. The inhibition of IstTx on the TREK-1 channel under several pH solutions was determined. A pH of 5.9 could significantly potentiate the inhibition of IstTx against the TREK-1 channel (Figure 6C). However, another two solutions (pH 6.9 and pH 6.4) showed little effect on the potentiation (Figure 6C). Under pH 5.9, IstTx inhibited the TREK-1 channel in a concentration-dependent manner, with an IC_50_ of 4.60 ± 0.04 μM, and the inhibitory efficacy was increased compared to that under pH 7.4 (Figure 6D,E). Residue His 142 in the TREK-1 channel was presumed to play an essential role in channel protonation [34]. Two mutants (H142S and H142K) of the TREK-1 channel were obtained, representing protonation non-ability and the protonated state of residue His142, respectively. However, IstTx displayed no significant difference on either mutant compared to that on a wild-type (WT) TREK-1 channel (Figure 6F), suggesting that His142 in the TREK-1 channel exhibited little effect against the affinity of IstTx acting on the channel. Mutation L304W dramatically reduced the affinity of fluoxetine to the TREK-1 channel (Figure 6G). However, IstTx displayed a similar inhibitory efficacy on WT and mutant L304W of the channel (Figure 6H). Therefore, we indicated that acidification could enhance the inhibition of IstTx on the TREK-1 channel, and our results indicated that IstTx may inhibit the channel in a different mechanism from that of fluoxetine.

### 2.6. Molecular Docking of the Peptide IstTx Interacting with the TREK-1 Channel

A virtual molecular docking was carried out by Discovery studio using the predicted IstTx structure (Figure 1C) and the reported TREK-1 channel Cyro-EM structure (PDB ID: 6cq6) [36]. A total of 58 poses were generated by the molecular docking without setting binding sites, 15 of which showed that peptide IstTx might bonded to the extracellular region of the TREK-1 channel (Figure 7A). Considering that the peptide IstTx was quickly washed off when perfusing from the outside of the cells (Figure 5C), the extracellular region of the TREK-1 channel was presumed to be the IstTx-bonded site. Three poses with a high Z-score were assumed to reflect the possible interaction between the peptide IstTx and the TREK-1 channel (Figure 7B–D). A salt bridge plays an important role in a peptide interacting with its receptor [35,37,38]. We analyzed several salt bridges that might link the peptide IstTx with the TREK-1 channel. Residue Glu17 in IstTx might interact with Lys156 in the TREK-1 channel with an ionic bond (Figure 7E), and Arg50 of IstTx might link to Glu153 of the channel (Figure 7F). The interaction between Arg82 of IstTx and Glu101 of the channel was observed (Figure 7G). Therefore, the possible mechanism of the peptide IstTx binding to the TREK-1 channel was revealed by virtual molecular docking.

## 3. Discussion

Animals’ venoms play a critical role in their survival and competition in a complicated environment. Animals take advantage of their toxins in many ways [39,40,41,42,43]. For example, frogs protect their bare skin with multiple antimicrobial peptides in order to subsist in a damp habitat [44,45,46], and horseflies release a number of anticoagulant peptides when they suck blood [47]. Increasing evidence has shown that toxins acting on ion channels make an essential contribution to animals when hunting prey or defending against enemies, such as RhTx from centipedes targeting the TRPV1 channel [38], and scorpion α-toxins activating the sodium channel [37]. Notably, some peptides that are derived from animal toxins are presumed to be potent candidates for the treatment of some diseases, including NaSpTx family toxins for pain treatment [39] and Shk-186 for curing autoimmune diseases [48]. Consequently, exploring and analyzing the contents of animal toxins and their corresponding targets is indispensable for understanding the rules of animal survival and seeking effective drug candidates. The collection and purification of animal venoms is a direct way to attain some specific toxins. However, this is challenged by the paradox between collecting animal venoms and protecting animals. In this study, we selected a peptide from a cDNA library and obtained purified IstTx via recombinant expression and multiple purifications (Figure 1 and Figure 2). Screening a cDNA library greatly facilitates the analysis of the components of animal venoms, thus facilitating the understanding of animal behaviors and the exploration of medicinal drugs. By sequence alignment, we found that IstTx displayed a high homology with some peptides derived from other ticks (Figure 1B) and that IstTx also highly overlapped with the complement inhibitors CirpT1-4 [31] (Figure 1C), suggesting that IstTx might be involved in the complement system.

Sequence alignment also showed that IstTx shared a similar structure with the venom peptide MmKTx1 and U-scoloptoxin (16)-Sm2a, both of which are predicted by automated computational analysis. Since many toxins have been reported to act on ion channels and receptors that are also identified to involve many life processes, including pain sensation [49], we sought the targets of IstTx in ion channels. However, IstTx displayed no effect on all tested voltage-gated sodium, potassium, or calcium channels and was not targeted on a chloride channel (Ano1) or proton channel (Otop1) (Figure 4). Luckily, a K2P channel, TREK-1, was identified as a receptor for IstTx, and another three K2P channels were slightly inhibited by IstTx (Figure 3). These results indicated the selectivity of IstTx acting on the TREK-1 channel. The TREK-1 channel regulates resting membrane potential and relates to the central nervous system, and the channel can be gated by both physical and chemical stimuli, such as membrane stretching and anionic lipids [50]. The TREK-1 channel is proposed to be an important target of antidepression treatments, and the channel is inhibited by several potential antidepressants, norfluoxetine/fluoxetine [28,29]. In this study, we identified a novel peptide from *Ixodes scapularis* selectively inhibiting the TREK-1 channel (Figure 3). IstTx strongly acted on the TREK-1 channel when the channel was stimulated by AA (Figure 5C), which is different from other inhibitors. 

The molecular mechanism of the inhibition of TREK-1 channels by different antagonists were unrevealed through electrophysiological recording and Cryo-EM structure. For example, the binding sites of norfluoxetine/fluoxetine are located in the open fenestration pocket, and mutation of the key site Leu304 in the TREK-1 channel decreased the inhibition by fluoxetine [21] (Figure 6G). A small-molecules TKDC (N-(4-chlorophenyl)-N-(2-(3,4-dihydrosioquinolin-2(1H)-yl)-2-oxoethyl) methane sulfonamide) selectively targeted the extracellular allosteric cap of the TREK-1 channel and ruthenium amines were shown to interact with the keystone inhibitor sites above the selective filter of the TREK-1 channel, both obstructing the ion conduction pathway [51,52]. Spadin, an endogenous peptide derived from sortilin, exerted the potent inhibitory efficiency on the TREK-1 channel with the IC_50_ of the nanomolar range [53]. However, the mechanism of spadin blocking the TREK-1 channel was not established. IstTx might act on the TREK-1 channel in a different manner from fluoxetine (Figure 5B and Figure 6G,H), and the virtual molecular docking discovered the possible mechanism of IstTx interacting with the channel (Figure 7A–D). Several residues located in the extracellular domain of the TREK-1 channel might be crucial for IstTx binding to the TREK-1 channel, such as Glu101, Glu153, and Lys156 (Figure 7E–G). IstTx might block the extracellular ion pathway of the TREK-1 channel, which is similar to the inhibition of the TREK-1 channel by TKDC or ruthenium amines. However, the hypothesis needs more evidence from electrophysiological recording or Cryo-EM structure. TREK-1 channel inhibitors, such as TKDC and spadin, produced an antidepressant phenotype in animal tests [52,54]. IstTx, as a novel peptide acting on the TREK-1 channel, might serve as a promising drug template for treating depression in the future.

Many animal venoms display an acidic feature, and the acidic environment is critical to the stabilization of the toxins’ activity. Some reports have revealed that acidification potentiates the function of toxins on their receptors [35]. The efficacy of IstTx acting on the TREK-1 channel was enhanced under pH 5.9, and the channel was also regulated by extracellular protons [34] (Figure 6A–E), suggesting that IstTx synergized with protons when inhibiting the channel. Through its synergistic effect with protons, IstTx effectively exerts its activity. However, the mechanism needs to be clarified in our next exploration. The TREK-1 channel is also a mechanosensitive channel and is activated by membrane stretch, which makes it involved in pain perception [55]. The TREK-1 channel could heterodimerize with both the TREK-2 and the TRESK channels [56,57]. We found that IstTx exerted a weak affinity to the TREK-2 and TRESK channels (Figure 3), and considering that IstTx might have obstructed the extracellular ion pathway of the TREK-1 channel (Figure 7), IstTx might display a tiny inhibition on these heterodimerizes. The TREK-1 channel, as a background potassium channel correlated with the TRESK channel, played a critical role in neuronal excitability [58]. With IstTx, ticks might increase pain sensation by inhibiting the TREK-1 channel, which is thus conducive to defending against their predators.

In summary, we identified a novel peptide, IstTx, from the *Ixodes scapularis* cDNA library and obtained the peptide via recombinant expression and purification. Patch clamp recording showed that it selectively blocked the potassium channel TREK-1. Moreover, IstTx also acted on the activated state of the TREK-1 channel induced by AA and might bind to different sites on the channel from that of fluoxetine. Interestingly, extracellular acidification enhanced the affinity of IstTx to the channels, and the synergistic effect with protons may effectively exert its function. Therefore, our finding indicated that IstTx displays a potent inhibitory efficacy on the TREK-1 channel, which provides a novel probe to explore the channel and offers a new strategy for analyzing the relationship of animal toxins with their receptors.

## 4. Materials and Methods

### 4.1. Sequence Alignment and Homologous Modeling

The sequence of IstTx was obtained from the cDNA library of the salivary gland of the tick *Ixodes scapularis*. Via the Basic Local Alignment Search Tool (BLAST, https://blast.ncbi.nlm.nih.gov (accessed on 26 July 2024)), the protein sequences of CirpT1-4, MmKTx1, and Sm2a were achieved. Sequence alignment was carried out by Muscle in Molecular Evolutionary Genetics Analysis (MEGA) version 10.1. The structure of IstTx was predicted in Swiss-model (https://swissmodel.expasy.org (accessed on 26 July 2024)). The structure of CirpT1 was obtained from Protein Data Bank (PDB ID: 6rpt, https://www.pdbus.org (accessed on 26 July 2024)). After removing the receptor and refining contents in spatial structure, comparison of IstTx with CirpT1 was performed in PyMOL molecular graphics system version 2.4.0. 

### 4.2. Recombinant Expression and Purification of IstTx

IstTx was inserted into the plasmid of pET-32a (+), and the enterokinase recognition site (DDDDK) and 6 × His tag was also inserted into the plasmid, as shown in Figure 2A. After induction by 0.6 mM isopropyl-beta-D-thiogalactopyranoside (IPTG, Waltham, MA, USA, Cas # 367-93-1) for 12 h at 16 °C, BL21 (DE3) cells were collected and disrupted by ultrasonication. The supernatant was performed to a Ni-NTA column to purify the recombinant protein, which was then digested to release the IstTx by enterokinase (Cas # 9014-74-8) for 16 h at 37 °C. The purified protein was checked via sodium dodecyl sulfate-polyacrylamide gel electrophoresis (SDS-PAGE). IstTx was purified via Sephadex G-75 (Amersham Bioscience, Amersham, UK, Cas # 37224-29-6) at a flow rate of 2 mL/min, followed by a purification by RP-HPLC C4 column (America Sepax, 300 Å, 5 µm, 4.6 mm × 250 mm, 110045-10025) at a flow rate of 1 mL/min at room temperature. The molecular weight of IstTx was identified by MALDI-TOF-MS. After lyophilization, IstTx was kept at −20 °C until use.

### 4.3. Cell Culture and Plasmids Transfection

HEK293T cells were cultured in Dulbecco’s modified Eagle’s medium (DMEM) with 10% fetal bovine serum, 1% penicillin, and streptomycin in a 5% CO_2_ incubator, as performed previously [59]. Mutants of the TREK-1 channel were obtained by using a site-directed mutagenesis kit (Vazyme Cat. C214-01, Nanjing, China). WT or mutants’ plasmids of various channels were transiently transfected into cells with the reagent polyJet (SignaGen, Waltham, MA, USA) or Lipofectamine 2000 (Invitrogen, Waltham, MA, USA). Briefly, cells were planted in fresh DMEM before transfection. The channels’ plasmid was incubated with the transfection reagent for 20 min, and a plasmid expressing enhanced green fluorescent protein (eGFP) was also transfected into the cells to identify the transfected cells at the same time. After 6–8 h, cells were transferred into complete DMEM, and after 15–18 h of transfection, whole cell patch clamp recording was carried out to check the currents. Plasmids of Nav1.3, Nav1.4, Nav1.5, Nav1.7, Kv2.1, Kv4.1, Cav3.1, and BK channel were obtained from our lab. Cells stably expressing Kv1.5 were donated by Dr. Tang Qiang of Huazhong University of Science and Technology. Plasmid of ANO1 channel came from Pro. Zhang Hailin of Hebei Medical University. Plasmids of TREK-1 and TREK-2 channels were donated by Pro. Li Yang of Shanghai Institute of Materia Medica, Chinese Academy of Sciences. Plasmids of pcDNA 3.1 containing Otop1 channel, KCNK3, KCNK9, and KCNK18 were constructed in our lab.

### 4.4. Electrophysiology

After 15–18 h of transfection, the transfected cells were checked for currents using whole cell patch clamp recording with an EPC-10 amplifier and PatchMaster software (v.2x92) or a MultiClamp 700B amplifier and pCLAMP software (HEKA, v.10), as performed previously [60]. Patch electrodes were polished with a resistance of 3–5 MΩ for recording. For Nav channel recording, bath solution contained (mM) 140 NaCl, 3 KCl, 1 CaCl_2_, 1 MgCl_2_, 10 HEPES, pH 7.4, and pipette solution contained (mM) 140 CsF, 10 NaCl, 1 EGTA, 10 HEPES, pH 7.2. For Kv channel recording, bath solution contained (mM) 140 NaCl, 3 KCl, 2 CaCl_2_, 1.5 MgCl_2_, 10 HEPES, 10 Glucose, pH 7.4, and pipette solution contained (mM) 145 KCl, 1 MgCl_2_, 10 HEPES, 5 EGTA, pH 7.2. For Cav3.1 channel recording, bath solution contained (mM) 20 CsCl, 140 TEA-Cl, 5 BaCl_2_, 30 Glucose, 25 HEPES, pH 7.2, and pipette solution contained (mM) 110 CsCl_2_, 4 MgATP, 0.3 Na_2_GTP, 10 EGTA, 25 HEPES, pH 7.4. For BK channel recording, bath and pipette solutions contained (mM) 140 KMeSO_3_, 5 NaCl, 10 HEPES, pH 7.4. For Otop1 channel recording, pipette solution contained (mM) 120 Cs-aspartate, 15 CsCl, 2 Mg-ATP, 5 EGTA, 2.4 CaCl_2_, 10 HEPES, pH 7.3, and bath solution contained (mM) 160 NMDG-Cl, 2 CaCl_2_, 10 mM buffer based on pH (HEPES for pH 8–7.4, HomoPIPES for pH 5.5–4). For Ano1 channel recording, the high Ca^2+^ pipette solution contained (mM): 146 CsCl, 2 MgCl_2_, 5 mM Ca^2+^-EGTA (free Ca^2+^ ∼20 µM), 10 Glucose, 10 HEPES, pH 7.3, and bath solution contained (mM) 140 NaCl, 5 KCl, 2 CaCl_2_, 1 MgCl_2_, 15 glucose, 10 HEPES, pH 7.4. For K2P channels recording, bath solution contained (mM) 140 NaCl, 5 KCl, 1 MgCl_2_, 2 CaCl_2_, 10 HEPES, 10 glucose, pH 7.4, and pipette solution contained (mM) 10 NaCl, 117 KCl, 2 MgCl_2_, 10 HEPES, 10 EGTA, pH 7.2. All regents were obtained commercially from Sigma. Effect-concentrations were fitted to a Hill equation.

### 4.5. Molecular Docking

Molecular docking was performed by Discovery Studio version 2019 as performed previously [40]. Briefly, predicted IstTx structure (Figure 1C) and reported Cryo-EM TREK-1 channel structure (PDB ID: 6cq6) were employed to molecular docking using ZDOCK 3.0.2 and RDOCK without setting key binding site. Angular step size was set as 6^◦^, 6.0 Å for RMSD cutoff and 9.0 Å for interface cutoff in all ZDOCK docking.

### 4.6. Data Analysis

The data in this study were performed with Igor Pro (v8), GraphPad Prism (v9), or OriginLab (v2022), and Adobe Illustrator (v2021). All statistical values were performed as means ± standard errors of the mean of n repeats of the experiments. All data were analyzed using statistics (*t* test or ANOVA) to identify significant or non-significant mean differences.

## Figures and Tables

**Figure 1 ijms-25-08377-f001:**
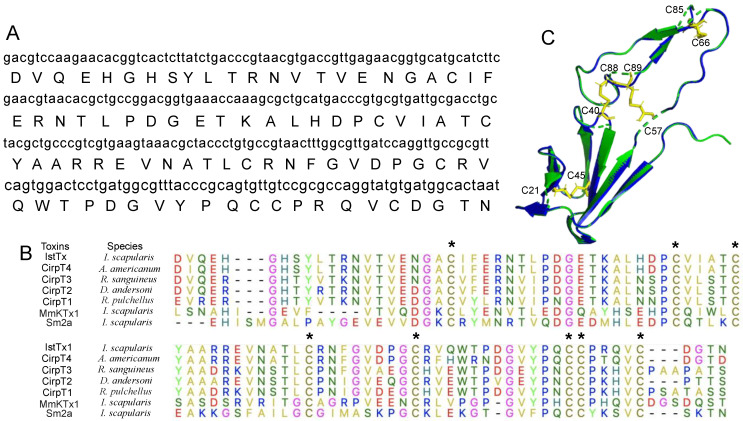
Structure of IstTx. (**A**) The cDNA sequence (lowercase letters) and amino acids (uppercase letters) of mature peptide IstTx. (**B**) Sequence alignment of IstTx with some toxins obtained from other ticks. Cysteines marked by *. (**C**) Comparison of spatial structures of complement inhibitor CirpT1 (isolated from PDB ID: 6rpt, blue) and IstTx (homology modeling based on CirpT1, green). The cysteines with their residue number in IstTx are labeled, and four intramolecular disulfide bonds labeled in yellow.

**Figure 2 ijms-25-08377-f002:**
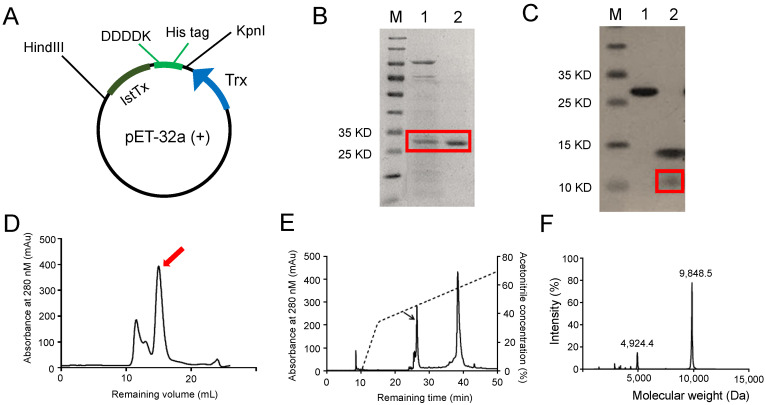
Recombinant expression and purification of peptide IstTx. (**A**) Sequence of IstTx was inserted into pET-32a (+) plasmid. Enterokinase recognition site (DDDDK) and 6×His tag also inserted into plasmid. (**B**) SDS-PAGE checking purification of recombinant IstTx by Ni-NTA column. Recombinant IstTx labeled with a red box. M: marker; line 1: sample in washing buffer; line 2: sample in elution buffer. (**C**) SDS-PAGE checking release of mature peptide IstTx by enterokinase. Mature peptide IstTx labeled with a red box. M: marker; line 1: sample before treatment by enterokinase; line 2: sample after treatment by enterokinase. (**D**,**E**) Purification of mature peptide IstTx via Sephadex G-75 (**D**) and RP-HPLC (**E**). Peaks containing peptide IstTx labeled with an arrow. (**F**) Identification of peptide IstTx by MALDI-TOF mass spectrometry.

**Figure 3 ijms-25-08377-f003:**
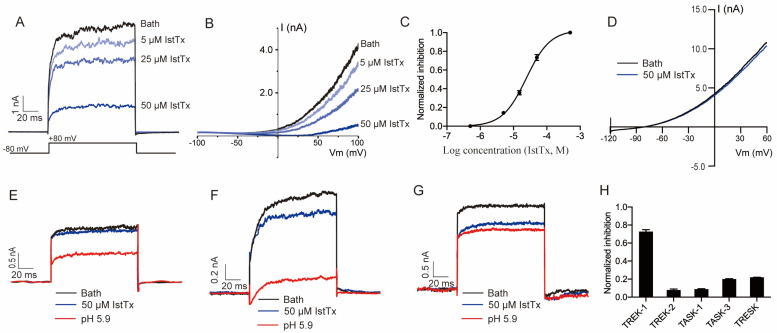
Inhibitory effects of peptide IstTx on K2P channels. (**A**) IstTx inhibited the current of TREK-1 channel, evoked by stable potential of +80 mV. Cell membrane held in −80 mV. (**B**) IstTx inhibited current of TREK-1 channel evoked by ramp potential of −100 mV to +100 mV in 100 ms. (**C**) Concentration-response displaying inhibition of IstTx on TREK-1 channel. Data (n = 5–8) fit by Hill equation. (**D**) IstTx showed no activity on TREK-2 channel evoked by ramp potential of −120 mV to +160 mV in 200 ms. (**E**–**G**) IstTx inhibited currents of K2P channels TASK-1 (**E**), TASK-3 (**F**), and TRESK (**G**) evoked by +80 mV. Cell membrane held in −80 mV. (**H**) Statistical analysis of 50 μM IstTx inhibiting several K2P channels, data coming from (**A**,**D**–**G**).

**Figure 4 ijms-25-08377-f004:**
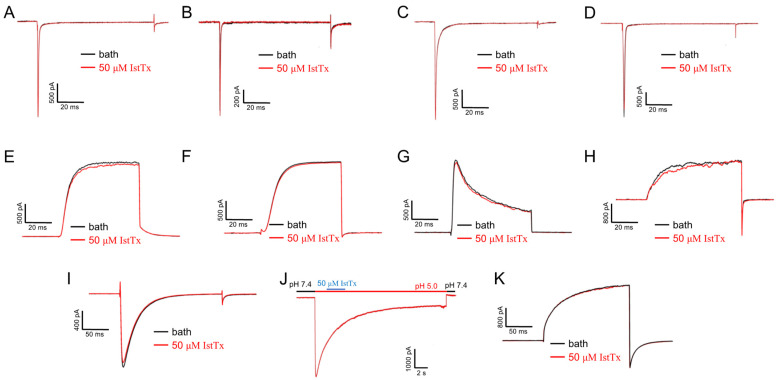
Effects of IstTx on other channels. (**A**–**D**) IstTx did not inhibit voltage-gated sodium channel Nav1.3 (**A**)**,** Nav1.4 (**B**), Nav1.5 (**C**), and Nav1.7 (**D**). Currents of these channels evoked by −10 mV and cell membranes held in −80 mV. (**E**–**G**) IstTx did not inhibit voltage-gated potassium channel Kv1.5 (**E**), Kv2.1 (**F**), and Kv4.1 (**G**). Currents of these channels evoked by 0 mV, and cell membranes held in −80 mV. (**H**) IstTx displayed no inhibitory efficacy on BK channel co-expression with β1 subunit. Current evoked by +120 mV, and cell membranes held in −80 mV. (**I**) IstTx displayed little effect on voltage-gated calcium channel Cav3.3. Current evoked by −10 mV, and cell membranes held in −80 mV. (**J**) IstTx did not act on proton-activated proton channel Otop1. Cell membranes held in −60 mV. (**K**) IstTx exhibited no inhibitory efficacy on calcium-activated chloride channel Ano1. Current evoked by +80 mV and cell membranes held in −80 mV.

**Figure 5 ijms-25-08377-f005:**
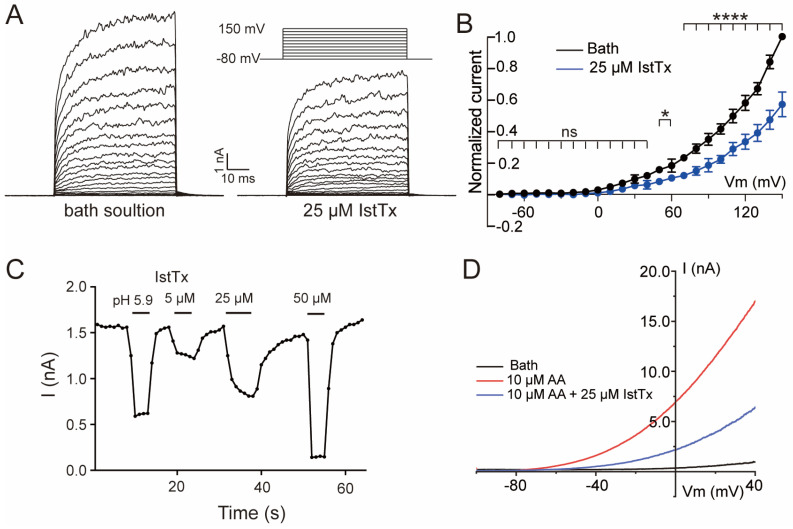
Some features of IstTx inhibiting TREK-1 channel. (**A**) IstTx inhibited TREK-1 channel in voltage-independent manner. Membrane potentials performed. (**B**) Statistical analysis of voltage dependence of IstTx inhibiting TREK-1 channel; data coming from (**A**). Significant differences calculated for each voltage step; ****, *, and ns indicate *p* ≤ 0.0001, *p* ≤ 0.05, and *p* > 0.05, respectively; n = 3. (**C**) Kinetics of IstTx acting on TREK-1 channel. Different concentrations of peptide IstTx and pH 5.9 perfused to channel, and current of TREK-1 channel evoked in +80 mV. (**D**) IstTx inhibited current of TREK-1 channel evoked by AA. Ramp recording performed from −100 mV to +40 mV in 100 ms.

**Figure 6 ijms-25-08377-f006:**
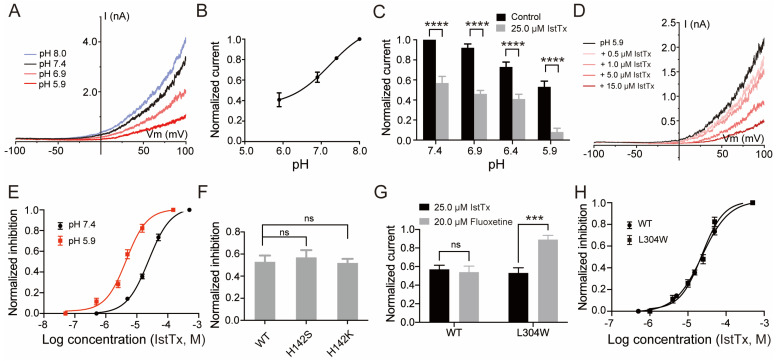
Acidification enhanced inhibitory efficacy of IstTx on TREK-1 channel. (**A**,**B**) Representative current traces (**A**) and effect-concentration (**B**) of effect of pH on TREK-1 channel. Data (n = 4–6 cells) were fit to Hill equation. (**C**) Effect of different pH alone or co-perfusion with IstTx on TREK-1 channel, n = 4–6 cells. (**D**) Representative current traces of different concentrations of IstTx on TREK-1 channel co-perfusion with pH 5.9. (**E**) Effect-concentration of IstTx on TREK-1 channel in neutral or pH 5.9 solution fit to Hill equation (n = 5–8 cells). (**F**) Inhibition of 25 μM IstTx on WT, mutant H142S, and H142K of TREK-1 channel (n = 5–7 cells). (**G**) Inhibition of 25 μM IstTx or 20 μM fluoxetine on WT and mutant L304W of TREK-1 channel (n = 5–8 cells). (**H**) Effect-concentration of IstTx on WT and mutant L304W of TREK-1 channel. Data (n = 5–8 cells) were fit to Hill equation; ****, ***, and ns indicate *p* ≤ 0.0001, *p* ≤ 0.001, and *p* > 0.05, respectively.

**Figure 7 ijms-25-08377-f007:**
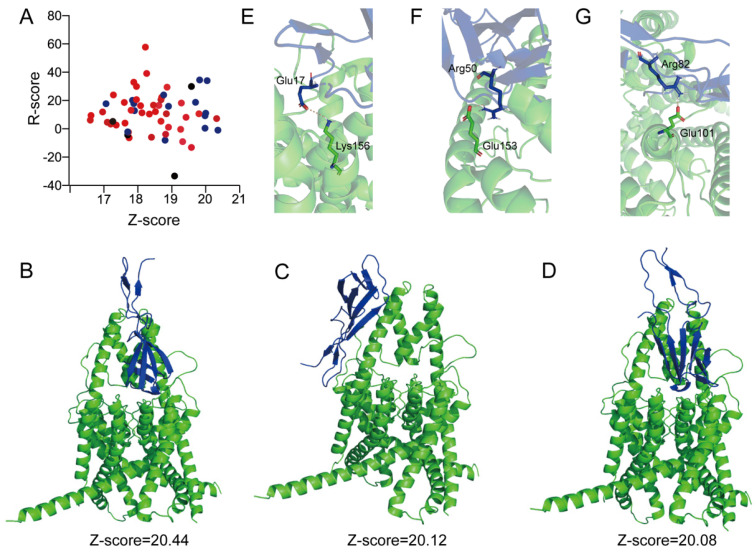
Molecular docking of peptide IstTx binding to TREK-1 channel. (**A**) R-score and Z-score of all 58 poses of peptide IstTx binding to TREK-1 channel. Extracellular, intracellular, and transmembrane locations of IstTx represented by blue, red, and black dots, respectively. (**B**–**D**) Representative poses of IstTx binding to TREK-1 channel by molecular docking. Z-score of each pose labelled. Peptide IstTx and TREK-1 channel colored blue and green, respectively. (**E**–**G**) Possible salt bridges of critical residues in peptide IstTx interacting with TREK-1 channel. Critical residues of peptide and IstTx and TREK-1 channel shown in stick. Peptide IstTx and TREK-1 channel colored blue and green, respectively. Salt bridge wheat-colored.

## Data Availability

The raw data supporting the conclusions of this article will be made available by the authors on request.

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
