# Peer review of "Tick-Derived Peptide Blocks Potassium Channel TREK-1"

_ijms, 2024, doi:10.3390/ijms25158377_

Round 1

Reviewer 1 Report

Comments and Suggestions for Authors

In this article, authors found and showed an attractive peptide inhibitor IstTx for TREK-1 channels, which belong to two-pore K+ channels involved in the regulation of resting potential of neurons, secretary cells, and others.

Inhibitors of TREK-1 channels have been focused on as important therapeutic compounds in several types of diseases, in particular, depression (see Mathie A et al., Annual Rev. Pharmacol. and Toxicol. (2021); Two-Pore Domain Potassium Channels as Drug Targets: Anesthesia and Beyond.). The review showed with 3D analysis of the TREK-1 channels that there are at least three binding sites for inhibition of TREK-1 activities like in the case of IstTx. 

Three points should be added to the discussion section as follows,

#1 Authors would emphasize that the IstTX and related peptides would be attractive for one of the therapeutic TREK-1 channels.

#2 Although authors partly refereed the possible binding sites of IstTx and compared its characteristics to that of fluoxetine, more detailed investigation should be included in the paper based on the review of Mathie A et al. mentioned above; there are at least three independent binding sites for the inhibition of TREK-1 channels.

#3 TREK-1 channel distributions are already demonstrated in several papers. Where are the most probable targets for TREK-1 in the body in the case of IstTx? Interpretation would be helpful to the readers to understand the site(s) of action of Tick and other animals (due to inhibition of TREK-1 channels).

Author Response

1.In summary: Thank you very much for taking the time to review our manuscript. As you suggested, we have aded the discussion.

Three points should be added to the discussion section as follows,

Comment #1 Authors would emphasize that the IstTX and related peptides would be attractive for one of the therapeutic TREK-1 channels.

Response 1: Thank you. As we know, small molecule inhibiting TREK-1 channel exerted an antidepressant phenotype in animal model. IstTx, as a novel peptide acting on TREK-1 channel, might serve as a promising drug template for treating depression in the future. We emphasized the attractive point of IstTX and related peptides in discussion. We added the statement in discussion as “TREK-1 channel inhibitors, such as TKDC and spadin, produced an antidepressant phenotype in animal test52,54. IstTx, as a novel peptide acting on TREK-1 channel, might serve as a promising drug template for treating depression in the future.”

 Comment #2 Although authors partly refereed the possible binding sites of IstTx and compared its characteristics to that of fluoxetine, more detailed investigation should be included in the paper based on the review of Mathie A et al. mentioned above; there are at least three independent binding sites for the inhibition of TREK-1 channels.

Response 2: Thank you. As you suggested, we discussed the independent binding sites for the inhibition of TREK-1 channels based on the pepar of Mathie A et al. and the molecular docking showed that IstTx might bind to the extracellular sites of TREK-1 channel and block the extracellular ion pathway of TREK-1 channel, which was similar to the inhibition of TREK-1 channel by TKDC or ruthenium amines.

Comment #3 TREK-1 channel distributions are already demonstrated in several papers. Where are the most probable targets for TREK-1 in the body in the case of IstTx? Interpretation would be helpful to the readers to understand the site(s) of action of Tick and other animals (due to inhibition of TREK-1 channels).

Response 3:Thank you. As a background potassium channel, TREK-1 channel plays an important role in neuronal excitability. “With IstTx inhibiting TREK-1 channel, ticks might increase the pain sensation, which is thus conducive to defensing against their predators.” We made the discussion to explain the site of action of Tick through IstTx inhibiting TREK-1 channel.

Reviewer 2 Report

Comments and Suggestions for Authors

The manuscript by Du et al. describes a novel peptide toxin inhibitor of the TREK-1 channel. The Authors identified the toxin from a cDNA library, purified it and performed its functional characterization. In general, the manuscript is well written, there are a few stylistical and typographical errors which can be corrected after a thorough rereading of the text. The experiments are mostly well described and performed, although there are some points which should be paid attention to. A major question is how useful the peptide will be for in vivo studies, given its relative low affinity (~20 micromolar IC50).

Major questions:

Spadin, an endogenous inhibitor of TREK-1 is completely omitted from the paper, except for the fact that it has no effect on activated TREK-1. This statement itself is controversial (other groups say the effect is more pronounced on preactivated TREK-1 channel). However, given the fact that spadin has a higer affinity than IsTx, discussing spadin would be important in the paper. What would the potential advantages of using IsTX be compared to spadin, given its lower affinity?

TREK-1 channel can heterodimerize with both TREK-2 and TRESK subunits  (https://doi.org/10.1073/pnas.1522748113https://doi.org/10.1073/pnas.1522459113,https://doi.org/10.1074/jbc.M116.719039,  https://doi.org/10.1016/j.neuron.2018.11.039,https://doi.org/10.1074/jbc.RA120.014125) . The heterodimers have different pharmacological properties than the homodimers, perhaps also different sensitivity to IsTX?. If not tested experimentally, at least should be discussed.

Figure 3: The sample sizes of the data are too small for ANOVA comparison (n=3 per channel). Showing % inhibition for channel would be better.

The molecular docking studies in their current form do not enhance the manuscript. Why weren’t any of the predicted binding residues in either the peptide or the channel mutated to test the potential binding location experimentally??

Discussion:

Fluoextine is a clinically used drug, but its beneficial effect has not been shown to be mediated by TREK-1.

Inhibition of TREK-1 would lead to increased pain sensation, not decreased as suggested in the text. Please rewrite

Minor comments:

Introduction:

TREK-1 is expressed in the sensory neurons of the dorsal root and trigeminal ganglia yet is not mentioned in the Introduction. This is puzzling given that is mentioned in the discussion.

TREK-1 is also expressed in the heart, with quite a lot of literature. This should be at least briefly mentioned in the introduction.

The pulmonary circulation is mentioned, but not some recent new studies highlighting the functional importance of the TREK-1 channel in the pulmonary circulation (https://doi.org/10.1111/bph.16426 , https://doi.org/10.3389/fcvm.2024.1343804.

Results: line 91- assimilated , should be estimated/predicted instead

Figure 3: please recheck the figure legends, there are some errors regarding the labelling of the panels and the text.

Line 170: Stepwise voltage protocol would be more clear instead of stable-state activated patch

Line 185: Distant, should be distinct instead

Figure 5d: How does the degree of TREK-1 inhibition in AA+IstTx compare to inhibition without AA? Is there any difference?

Line 206: The efficacy of the inhibition did not increase 5-fold in low pH, the potency of the compound increased 5-fold (22 micromolar to 4.6 micromolar). Please correct

Line 207: perfumed: Please check this sentence for accuracy

Figure 6f: I see no significant differences between the mutants, please rewrite the text to imply the same

Line 214: The referenced experiment shows pH washoff not fluoextine washoff. Please check and correct.

Comments on the Quality of English Language

Please recheck the manuscript, I have highlighted multiple errors of the text that make understanding the sentence much harder.

Author Response

In summary.

Thank you very much for taking the time to review our manuscript. Please find the detailed responses below and the corresponding revisions/corrections highlighted/in track changes in the re-submitted files.

Comment: The manuscript by Du et al. describes a novel peptide toxin inhibitor of the TREK-1 channel. The Authors identified the toxin from a cDNA library, purified it and performed its functional characterization. In general, the manuscript is well written, there are a few stylistical and typographical errors which can be corrected after a thorough rereading of the text. The experiments are mostly well described and performed, although there are some points which should be paid attention to. A major question is how useful the peptide will be for in vivo studies, given its relative low affinity (~20 micromolar IC50).

As you suggested, we made the thorough reading of the manuscript and corrected the stylistical and typographical errors. In this paper, we obtained the peptide IstTx by screening cDNA library and found IstTx electively inhibited TREK-1 channel. Importantly, many animal venoms display an acidic feature, and the acidic environment is critical to the stabilization of toxins activity. Under pH 5.9, IstTx inhibited the TREK-1 channel with an IC50 of 4.60 ± 0.04 μM. Thus, through its synergistic effect with protons, IstTx might effectively exert its activity and increase the pain sensation, which is thus conducive to defensing against their predators. On the other hand, “IstTx, as a novel peptide acting on TREK-1 channel, might serve as a promising drug template for treating depression in the future.” We made a discussion in discussion part.

Major questions:

Comment 1: Spadin, an endogenous inhibitor of TREK-1 is completely omitted from the paper, except for the fact that it has no effect on activated TREK-1. This statement itself is controversial (other groups say the effect is more pronounced on preactivated TREK-1 channel). However, given the fact that spadin has a higer affinity than IsTx, discussing spadin would be important in the paper. What would the potential advantages of using IsTX be compared to spadin, given its lower affinity?

Response 1:Thank you. We discussed the mechanism of different antagonists acting on TREK-1 channel, including norfluoxetine/fluoxetine, TKDC and ruthenium amines. However, the molecular mechanism of spadin inhibiting TREK-1 channel was unclear. Spadin, derived from the neurotensin 3 receptor propeptide sortilin, showed a potential novel treatment for certain forms of depression. Based on our result, IstTx, as a novel peptide derived from tick venom, might act on the TREK-1 channel in a different manner from fluoxetine (Figure 5B and 6G, H), but might block the extracellular ion pathway of TREK-1 channel, which was similar to the inhibition of TREK-1 channel by TKDC or ruthenium amines (Figure 7E-G). IstTx, as a novel peptide acting on TREK-1 channel, might serve as a promising drug template for treating depression in the future.

Comment 2: TREK-1 channel can heterodimerize with both TREK-2 and TRESK subunits (https://doi.org/10.1073/pnas.1522748113,https://doi.org/10.1073/pnas.1522459113,https://doi.org/10.1074/jbc.M116.719039,https://doi.org/10.1016/j.neuron.2018.11.039,https://doi.org/10.1074/jbc.RA120.014125) . The heterodimers have different pharmacological properties than the homodimers, perhaps also different sensitivity to IsTX?. If not tested experimentally, at least should be discussed.

Response 2: Thank you. In this paper, we found that IstTx exerted a weak affinity to TREK-2 and TRESK channel (Figure 3) and considering that IstTx might obstructed the extracellular ion pathway of TREK-1 channel (Figure 7), IstTx might display tiny inhibition on these heterodimerizes.As you suggested, we have made the discussion.

Comment 3: Figure 3: The sample sizes of the data are too small for ANOVA comparison (n=3 per channel). Showing % inhibition for channel would be better.

Response 3: We have showed normalized inhibition for channel in figure 3H as you suggested.

Comment 4: The molecular docking studies in their current form do not enhance the manuscript. Why weren’t any of the predicted binding residues in either the peptide or the channel mutated to test the potential binding location experimentally??

Response 4: In this paper, we performed the molecular docking to explore the possible binding sites of IstTx acting on TREK-1 channel and three slat bridges between IstTx and TREK-1 channel were presumed to link the peptide and the receptor (Figure 7). We discussed three binding sites for inhibition of TREK-1 activities, such as a small molecules TKDC and ruthenium amines. IstTx might block the extracellular ion pathway of TREK-1 channel, which was similar to the inhibition of TREK-1 channel by TKDC or ruthenium amines.

Discussion:

Comment 5: Fluoextine is a clinically used drug, but its beneficial effect has not been shown to be mediated by TREK-1.

Response 5:Thank you, we have changed the statement as “The TREK-1 channel is proposed to be an important target of antidepression, and the channel is inhibited by several potential antidepressants, norfluoxetine/fluoxetine.”

Comment 6: Inhibition of TREK-1 would lead to increased pain sensation, not decreased as suggested in the text. Please rewrite

Response 6: Thank you. We have changed the sentence as “with IstTx, ticks might increase the pain sensation by inhibiting TREK-1 channel, which is thus conducive to defensing against their predators”.

Minor comments:

Introduction:

Comment 1: TREK-1 is expressed in the sensory neurons of the dorsal root and trigeminal ganglia yet is not mentioned in the Introduction. This is puzzling given that is mentioned in the discussion.

Response 1: We have added the statement in the introduction.

Comment 2: TREK-1 is also expressed in the heart, with quite a lot of literature. This should be at least briefly mentioned in the introduction.

Response 2: We have briefly mentioned "the heart " in the introduction.

Comment 3: The pulmonary circulation is mentioned, but not some recent new studies highlighting the functional importance of the TREK-1 channel in the pulmonary circulation (https://doi.org/10.1111/bph.16426 , https://doi.org/10.3389/fcvm.2024.1343804) .

Response 3: We have updated the reference in the revised manuscript.

Comment 4:Results: line 91- assimilated , should be estimated/predicted instead

Response 4: We have addressed the issue in the revised manuscript.

Comment 5:Figure 3: please recheck the figure legends, there are some errors regarding the labelling of the panels and the text.

Response 5: We have addressed the issue in the revised manuscript.

Comment 6: Line 170: Stepwise voltage protocol would be more clear instead of stable-state activated patch

Response 6: As you suggested, we have addressed the issue in the  revised manuscript.

Comment 7: Line 185: Distant, should be distinct instead

Response 7: As you suggested, we have addressed the issue in the  revised manuscript.We have addressed the issue.

Comment 8: Figure 5d: How does the degree of TREK-1 inhibition in AA+IstTx compare to inhibition without AA? Is there any difference?

Response 8: "IstTx at 25 μM inhibited almost half current of the TREK-1 channel when the channel was pre-simulated by AA, and the inhibitory efficacy was similar to that of the channel when activated by voltage without AA". We have added the statement in the result.

 Comment 9: Line 206: The efficacy of the inhibition did not increase 5-fold in low pH, the potency of the compound increased 5-fold (22 micromolar to 4.6 micromolar). Please correct

Response 9: We have addressed the issue in the revised manuscript.

Comment 10: Line 207: perfumed: Please check this sentence for accuracy.

Response 10: We have addressed the issue in the revised manuscript.

Comment 11: Figure 6f: I see no significant differences between the mutants, please rewrite the text to imply the same

Response 11: We have addressed the issue in the revised manuscript.

Comment 12: Line 214: The referenced experiment shows pH washoff not fluoextine washoff. Please check and correct.

Response 12:  Thank you. We have removed the sentence.

Comments on the Quality of English Language

Please recheck the manuscript, I have highlighted multiple errors of the text that make understanding the sentence much harder.

Response: Thank you. We have rechecked the manuscript and addressed the issue as you suggested.

Round 2

Reviewer 2 Report

Comments and Suggestions for Authors

Thank you for addressing my comments.